# ZEROTOP: Zero-Shot Task-Oriented Semantic Parsing using Large Language Models

**Dheeraj Mekala\***
UC San Diego
dmekala@ucsd.edu

**Jason Wolfe♠**
OpenAI
jasonwolfe@openai.com

**Subhro Roy**
Microsoft Semantic Machines
subhro.roy@microsoft.com

## Abstract

We explore the use of large language models (LLMs) for zero-shot semantic parsing. Semantic parsing involves mapping natural language utterances to task-specific meaning representations. LLMs are generally trained on publicly available text and code and cannot be expected to directly generalize to domain-specific parsing tasks in a zero-shot setting. In this work, we propose ZEROTOP, a zero-shot task-oriented parsing method that decomposes semantic parsing problem into a set of abstractive and extractive question-answering (QA) problems. For each utterance, we prompt the LLM with questions corresponding to its top-level intent and a set of slots and use the LLM generations to construct the target meaning representation. We observe that current LLMs fail to detect unanswerable questions; and as a result, cannot handle questions corresponding to missing slots. We address this by fine-tuning a language model on public QA datasets using synthetic negative samples. Experimental results show that our QA-based decomposition paired with the fine-tuned LLM can zero-shot parse $\approx 16\%$ of utterances in the MTOP dataset.

## 1 Introduction

Large language models (LLMs) are trained on publicly available text (Raffel et al., 2020; Sanh et al., 2021; Brown et al., 2020) and code (Chen et al., 2021) and have been shown to attain reasonable zero-shot generalization on a diverse set of NLP tasks (Wang et al., 2019). However, they are not expected to generalize to domain-specific semantic parsing tasks in a similar way, where the inductive bias from pre-training is less helpful. In this work, we propose ZEROTOP that decomposes the semantic parsing task into one of answering a series of extractive and abstractive questions, corresponding

to its top-level intent and a set of relevant slots, and leverage the LLM's ability to zero-shot answer reading comprehension questions.

As illustrated in Figure 1, we cast top-level intent classification as an abstractive QA task. To address LLMs' bias towards predicting labels common in the pretraining data (Zhao et al., 2021), we propose to generate an intent description in an unconstrained manner and infer the intent label most similar to the generated description. We view slot value prediction as an extractive QA problem. Most utterances do not mention all the slots. It is therefore essential for the model to abstain from prediction when corresponding slots are not mentioned. Through our analyses, we observe that most LLMs frequently hallucinate text for missing slots with high confidence, resulting in poor performance. To address this, we fine-tune an LM on a collection of public QA datasets augmented with synthetic *unanswerable* samples. We call our trained model *Abstainer*, as it is capable of identifying unanswerable questions and abstaining from prediction. We hierarchically prompt for nested slots using the Abstainer, and infer nested intents if their corresponding slots are detected. We empirically show that this QA based decomposition of ZEROTOP is an effective way to leverage LLMs for domain specific semantic parsing, outperforming several strong baselines in the zero shot setting.

## 2 Related Work

LLMs are increasingly used for semantic parsing in low-data scenarios utilizing canonical representations (Shin et al., 2021; Yang et al., 2022), and prompt-tuning (Schucher et al., 2022; Drozdov et al., 2022). The closest work to ours is Zhao et al. (2022) where they decompose parsing into QA tasks. However, they assume access to some annotated data whereas we focus on a strict zero-shot setting where only the schema information is available along with some natural language prompts for

---

\* Work done during an internship at Microsoft Semantic Machines.

♠ Work done while at Microsoft Semantic Machines.

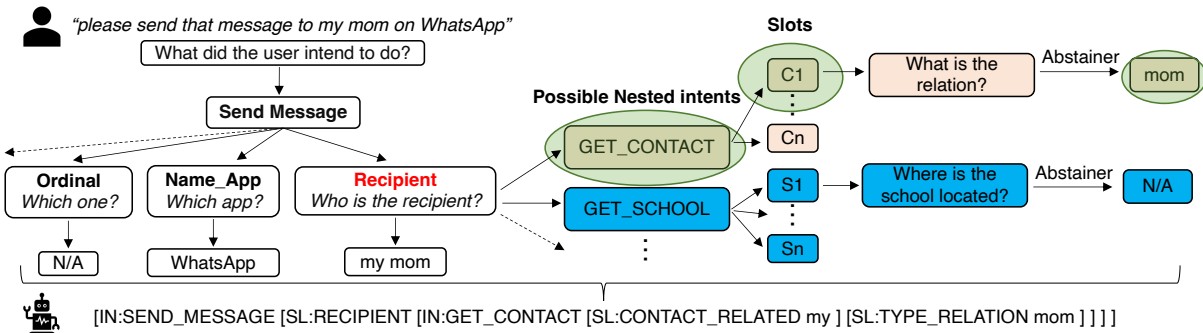

Figure 1: ZEROTOP hierarchically prompts LLMs for identifying intents, slots, and nested intents, and combines LLM generations to create meaning representation. First, we identify top-level intent. Next, we prompt for each slot. If a slot can accommodate nested intents (in red), we prompt for nested slots and infer nested intents.

schema entities. Our work is also related to approaches towards zero-shot dialog state tracking using LLMs (Gao et al., 2020; Lin et al., 2021a,b). Specifically, Lin et al. (2021a) uses an Abstainer to handle missing slots. Our method differs in that, we focus on semantic parsing where the Abstainer needs to be applied multiple times along with intent detection to create nested meaning representations.

## 3 ZEROTOP: Zero-Shot Task-Oriented Semantic Parsing

**Problem Formulation** We focus on task-oriented parsing with hierarchical intent-slot schema. Let $\mathcal{I} = \{\mathcal{I}_1, \mathcal{I}_2, \ldots, \mathcal{I}_n\}$ and $\mathcal{S} = \{\mathcal{S}_1, \mathcal{S}_2, \ldots, \mathcal{S}_m\}$ be the set of all possible top-level intents and slots respectively. Each intent $\mathcal{I}_j$ has a set of slots $\mathcal{S}^j = \{\mathcal{S}_1^j, \mathcal{S}_2^j, \ldots, \mathcal{S}_n^j\}$ that can be filled. Possible slots in an intent are represented by the intent-to-slot mapping I2S: $\mathcal{I} \to \mathcal{P}(\mathcal{S})$, where $\mathcal{P}(\cdot)$ is the powerset operator. Similarly, the inverse slot-to-intent mapping is represented by S2I: $\mathcal{S} \to \mathcal{I}$. The input in our setting consists of I2S and S2I, but no annotated data. ZEROTOP requires users to provide a question per slot $\mathcal{Q} = \{\mathcal{Q}_{\mathcal{S}_1}, \mathcal{Q}_{\mathcal{S}_2}, \ldots, \mathcal{Q}_{\mathcal{S}_k}\}$, that represents their purpose. In a real-life setting, this can be obtained from a domain developer.

**Unconstrained Generation for Zero-Shot Intent Classification** We view zero-shot intent classification as an abstractive QA problem. One intuitive way is to prime the LLM with a QA prompt and then constrain the generation to search over only valid intent labels (Shin et al., 2021). However, LLMs are known to be biased towards text sequences (Zhao et al., 2021) more common in pre-training data. For example, in the MTOP dataset, the T0-3B model predicts CREATE_CALL (*make*

---

**Algorithm 1:** ZEROTOP: Our proposed Zero-shot semantic parsing method.

**Input:** Set of intents $\mathcal{I}$, Set of slots $\mathcal{S}$, Slot questions $\mathcal{Q}$, intent-to-slot mapping I2S, slot-to-intent mapping S2I, slot-to-candidate-nested-intent mapping S2NI, Intent-model $\mathcal{M}_I$, Abstainer $\mathcal{M}_{abs}$, and Utterance $u$
**Output:** Predicted meaning representations MR
intent = $\mathcal{M}_I(u)$
slotValues = { }
**for** slot $\mathcal{S}_i \in$ I2S(intent) **do**
   slotValues[$\mathcal{S}_i$] = $\mathcal{M}_{abs}(u, \mathcal{Q}_{\mathcal{S}_i})$
   **for candidate N.intent** $\mathcal{I}_j \in$ S2NI($\mathcal{S}_i$) **do**
      **for slot** $\mathcal{S}_j \in$ I2S($\mathcal{I}_j$) **do**
         **if** $\mathcal{M}_{abs}$(slotValues[$\mathcal{S}_i$], $\mathcal{Q}_{\mathcal{S}_j}$) *is not NONE* **then**
            Update slotValues[$\mathcal{S}_i$] with nested intent $\mathcal{I}_j$, $\mathcal{M}_{abs}$(slotValues[$\mathcal{S}_i$], $\mathcal{Q}_{\mathcal{S}_j}$)
MR = Construct representation with intent, slotValues
**Return** MR

---

*call*) as intent for $92\%$ of the data in the *call* domain. Therefore, we propose to first generate an intent description in an unconstrained fashion by priming the LM with the following prompt.

Answer the following question depending on the context.
`context`: A user said, {*utterance*}.
`question`: What did the user intend to do?
`answer`:

Then, we choose the intent label that is most similar to generated answer using RoBERTa sentence similarity (Reimers and Gurevych, 2019). Unlike Zhao et al. (2022), our approach does not require enumerating choices in the prompt allowing us to handle a large number of intents and slots as found in datasets like MTOP.

**Leveraging QA datasets for Slot Value Prediction** Slot value prediction involves extracting phrases for a slot from the user utterance. We cast this as an extractive QA problem. All slots might

not be mentioned in an input utterance. For example, in the MTOP dataset, on average, only one-third of possible slots are mentioned per utterance. The QA model needs to abstain from prediction for such missing slots. To analyze the abstaining capability of pre-trained QA models, we consider a few top-performing zero-shot LLMs T0-3B, GPT-3, and Codex with their corresponding prompts and experiment on a 500 sample subset of unanswerable questions from the SQuAD dataset (Rajpurkar et al., 2018). We observe the accuracy of all models to be $< 5\%$ and notice that they frequently hallucinate and generate answers for unanswerable questions. In section 4, we also consider a log-likelihood-based threshold for abstaining and show that this threshold is difficult to tune using public QA datasets.

To address this challenge, we leverage multiple publicly available QA datasets[1] to train *Abstainer*, a QA model capable of abstaining from prediction. Specifically, we generate synthetic unanswerable training samples by modifying existing QA data, and train a QA model jointly on existing datasets and synthetic unanswerable questions. For every (question, answer, context) triplet, we generate synthetic unanswerable questions by either (1) removing the sentence containing the answer span from the context, or (2) randomly sampling a context that doesn't have the same question. After training the Abstainer, we prompt it for each slot with its corresponding question for slot value prediction, in the following format:

```
Answer the following question depending on the context.
context: A user said, {utterance}.
question: {slot question}
answer:
```

**Nested Intents**   To identify nested intents, we assume knowledge of candidate nested intents that can be accommodated by each slot, represented by the slot-to-candidate-nested-intent mapping S2NI: $\mathcal{S} \rightarrow \mathcal{P}(\mathcal{I})$. Our method assumes that depth of output representations is at most 4 i.e. nested intents cannot further have more nested intents. One intuitive way is to prompt the LLM for nested intent with the intent prediction prompt. However, our unconstrained generation-based intent model would predict many false positive nested intents. We instead use Abstainer to prompt for their respective slots. If any slot value is identified, we consider its corresponding intent via S2I to be present as well.

---

[1] The QA datasets details are mentioned in Appendix A.1

**ZEROTOP: Putting it all together**   The pseudo-code of ZEROTOP is mentioned in Algorithm-1. ZEROTOP employs a top-down, greedy prompting strategy, where we first prompt for intent and then, its respective slots. First, we obtain the top-level intent using the intent model. Based on the predicted intent, we prime the Abstainer for corresponding slots using their respective questions as prompts. For each identified slot value, we prompt the Abstainer for slots of candidate nested intents. We use the same prompt format for this step with the identified slot value now considered as the input utterance. Finally, we combine predicted intent, identified slot values, and nested intents to create the meaning representation.

## 4   Experiments

We experiment on the English language subset of MTOP (Li et al., 2021) dataset. MTOP is a multilingual task-oriented semantic parsing dataset comprising data from 6 languages and 11 domains. The test set has 4386 samples with 113 distinct intents and 74 slots. On average, each intent has 3.6 slots and 33% of possible slots are filled per utterance.

**Experiment Settings.**   We evaluate on the zero-shot setting, therefore we have no training data. We manually create questions for slots $\mathcal{Q}$ using one example per slot. For training Abstainer, we fine-tune T0-3B on the extractive and abstractive QA datasets for 1 epoch with a constant learning rate of $10^{-4}$. We use complete meaning representation match accuracy as the performance metric. More details in Appendix A.2.

**Baselines.**   We compare ZEROTOP with *constrained T0-3B, GPT-3* and *Codex* as intent models where they are primed with intent generation prompt and are constrained to search over valid intent labels. We also compare with *calibrated constrained T0-3B* (Zhao et al., 2021) whose logits are adjusted to counter LM biases. We consider an ablation of ZEROTOP where we assign intent labels based only on their similarity with user utterance using *RoBERTa sentence* transformer. **ZEROTOP-*Intent*** represents our proposed intent prediction method.

We compare with *constrained T0-3B, GPT-3, and Codex* as slot models as well, however, when primed with a question corresponding to a slot, the output is constrained to be either from the utterance or from their corresponding phrases indicating that question cannot be answered. We com-

| Intent Model | Accuracy(%) |
|---|---|
| T0-3B constrained | 34.02 |
| T0-3B constrained calibrated | 36.64 |
| GPT-3 constrained | 40.44 |
| Codex constrained | 48.02 |
| RoBERTa sentence | 47.14 |
| ZEROTOP-Intent | **49.58** |

Table 1: Top-level intent classification results.

| Intent Model | Slot Model | Acc(%) |
|---|---|---|
| GPT-3 constrained | GPT-3 constrained | 3.00[*] |
| Codex constrained | Codex constrained | 5.40[*] |
| T0-3B constrained | T0-3B constrained | 2.42 |
| | Abstainer | 11.81 |
| RoBERTa sentence | T0-3B constrained | 3.88 |
| | Abstainer | 12.90 |
| ZEROTOP-Intent | T0-3B constrained | 4.10 |
| | T0-3B constrained-MTQA | 4.63 |
| | T0-3B constrained-SEQZERO | 8.39 |
| | Abstainer-MTQA | 13.12 |
| | Abstainer-SEQZERO | 13.68 |
| | Abstainer | **15.89** |

Table 2: Complete meaning representation match evaluation. To limit API cost, we limit GPT-3 and Codex evaluation on a 500-example subset, and hence their results are not directly comparable.

pare with two kinds of prompting for slot values. *MTQA* (Zhao et al., 2022) propose prompting an LLM to identify filled slots and then prime for their respective values. *SEQZERO* (Yang et al., 2022) introduce prompting each slot sequentially and using the previously identified slot value for prompting the next one. *Abstainer* is our finetuned T0-3B that abstains from prediction. We prompt for each slot independently.

**Results and Discussion** From zero-shot intent classification results in Table 1, we observe that ZEROTOP-Intent performs significantly better than constrained T0-3B, GPT-3, and Codex. We found that constrained T0-3B is biased towards certain labels. For example, it predicts CREATE_CALL (*make call*), SEND_MESSAGE (*send message*), CREATE_REMINDER (*create reminder*), as intent for more than 90% of the data in *call*, *message*, *reminder* domains respectively. Our proposed unconstrained formulation lets the model freely express the intent and, computing similarity later with the intent labels addresses this bias.

As shown in Table 2, the combination of ZEROTOP-Intent and Abstainer demonstrates su-

perior performance than alternative combinations. We observe that T0-3B, GPT-3, and Codex fail to abstain frequently. The T0-3B model abstains only for 38% of unanswerable slot questions whereas Abstainer does for 89%. As a result, we observe a notable performance gain by plugging in Abstainer as the slot model for each intent model baseline. Moreover, we observe MTQA and SEQZERO prompting methods, which are originally proposed for custom-finetuned & few-shot models, offer little assistance in zero-shot settings. For instance, when applying MTQA prompting to the T0-3B model, we observe marginal improvements. This indicates that the pre-trained T0-3B is unable to accurately identify fillable slots, demonstrating the necessity of Abstainer. Similarly, while SEQZERO prompting improves the performance of T0-3B, its effectiveness remains significantly inferior to Abstainer. Finally, we see similar performance of MTQA and SEQZERO prompting with Abstainer. However, our approach of independently prompting for each slot outperforms them.

**Annotation Effort Analysis** We use 74 samples i.e. one per slot to design questions for slots. To analyze annotation effort, we train an utterance-to-meaning representation T5-3B (Raffel et al., 2020) parser using these 74 samples and compare it with our method. The match accuracy of T5-3B parser on the MTOP dataset is 8.19% and of ZEROTOP is 15.89%, justifying our annotation effort.

**Greedy vs Beam search** ZEROTOP follows a greedy strategy where we hierarchically prompt for top-level intent and for its corresponding slots. We compare it with the beam search strategy with beam size 3. Specifically, we consider 3 top-level intents and prompt for their corresponding slots, consider top-3 slot values for every slot and finally compute the best meaning representation based on their aggregated NLL scores. The NLL score of intent $\mathcal{I}_m$, its slots $\mathcal{S}_j \in \text{I2S}(\mathcal{I}_m)$, and their corresponding slot values $\text{slotValues}[\mathcal{S}_j]$ is computed:

$$\alpha \log p(\mathcal{I}_m) + (1-\alpha) \sum_{\mathcal{S}_j} \log p(\text{slotValues}[\mathcal{S}_j]|\mathcal{I}_m)$$

where $\alpha$ is tuned on a held-out validation set. Note that $p(\text{slotValues}|\mathcal{I}_m)$ is computed recursively for its nested intents. The complete match accuracy of the greedy prompting strategy on MTOP dataset is 15.89% and of beam search strategy is 16.86%. This demonstrates that beam search can improve performance with validation data. Without validation data and setting $\alpha$ to 0.5, performance drops

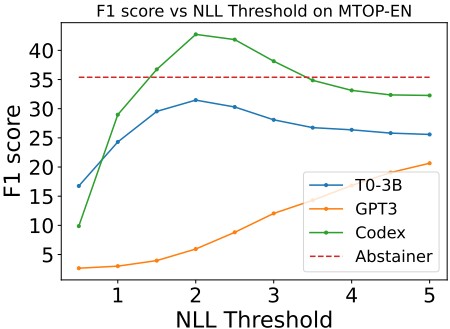

(a) F1 score vs NLL threshold on MTOP dataset

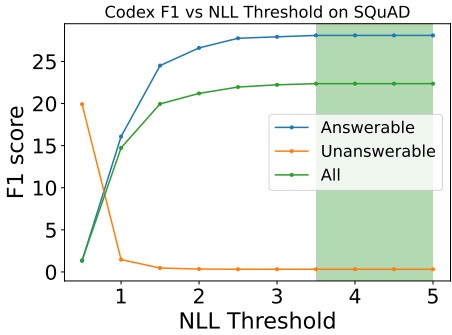

(b) F1 score vs NLL threshold on SQuAD dataset

Figure 2: We consider negative log-likelihood (NLL) as a confidence score and vary the threshold to abstain from prediction and plot F1 scores on the MTOP dataset. We show that this NLL threshold is difficult to tune using public QA datasets such as SQuAD as performance on answerable and unanswerable subsets is mutually exclusive.

to $12.36\%$ i.e. $3\%$ less than greedy. Therefore, we believe greedy prompting is a better choice.

**Confidence score-based Abstainer study**   We can alternatively have LLMs abstain from prediction based on a confidence score based threshold. We consider negative log likelihood (NLL) of the predicted slot value as the confidence score and abstain from prediction if it is greater than the threshold. We experiment on slot value prediction task with T0-3B, Codex, and GPT3 as LLMs and plot macro F1 scores for multiple NLL thresholds on a randomly sampled subset of 500 samples from MTOP dataset in Figure 2(a). Specifically, we consider the gold intent of each sample and prime LLM for extracting slot values for each slot of the gold intent. We consider F1 score as the metric due to the label imbalance across possible slot values. We present the F1-score of the Abstainer for reference. First, we observe that Abstainer is significantly better than T0-3B and GPT3 for all confidence thresholds. Second, we notice that there

is no threshold that consistently results in good performance for all LLMs, which implies that this has to be individually tuned for each LLM. Finally, we observe Codex performs better than Abstainer for some thresholds. As our problem setting includes no annotated data, we investigate whether we can infer the optimal threshold for Codex using public QA datasets. Specifically, we consider 500 answerable and 500 unanswerable QA pairs from SQuAD dataset and plot F1 scores with a range of confidence thresholds in Figure 2(b). We can observe that the performance on answerable and unanswerable subsets is mutually exclusive i.e. there is no threshold where the performance on both answerable and unanswerable subsets is high. The range of thresholds that result in the best performance on the whole set (highlighted in green) does not transfer to MTOP and is achieved at the cost of unanswerable set where the F1 score is less than $5\%$. Given the difficulty in tuning threshold and the API costs of Codex, we believe using Abstainer as the slot model to be a better choice.

# 5   Conclusion

In this paper, we propose ZEROTOP that decomposes semantic parsing into abstractive and extractive QA tasks. ZEROTOP identifies top-level intent by generating in an unconstrained fashion and inferring the intent label most similar to the generated description. We train Abstainer using public QA datasets, that is capable of identifying unanswerable questions and abstaining from prediction.

# 6   Limitations

ZEROTOP assumes that the meaning representations are of a limited depth i.e. nested intents cannot further have more nested intents and this is one of the limitations. Moreover, we also assume that it is possible to write natural questions corresponding to slots. A slot for which a natural question cannot be expressed, the LLM can't handle it without additional supervision. Finally, we believe there is a huge scope for improvement in the performance of LLMs and ZEROTOP in domain-specific tasks such as zero-shot semantic parsing and on the MTOP dataset.

# 7   Ethics Statement

This paper proposes a zero-shot semantic parsing method using large language models. The aim of the paper is to minimize the human effort in

annotation by leveraging language models. The output of our method is a meaning representation that doesn't contain any harmful content. Hence, we do not anticipate any major ethical concerns.

## 8 Acknowledgments

We thank the anonymous reviewers and our colleagues from Microsoft Semantic Machines, especially Hao Fang, Richard Shin, Adam Pauls, Matt Gardner, and Jason Eisner for their helpful feedback.

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

| Type | Dataset | # Samples |
|---|---|---|
| Extractive | Adversarial QA (Bartolo et al., 2020) | 36000 |
| | QA-SRL (He et al., 2015) | 8597 |
| | DuoRC (Saha et al., 2018) | 186089 |
| | ROPES (Lin et al., 2019) | 14000 |
| | SQuADv2 (Rajpurkar et al., 2018) | 150000 |
| | Quoref (Dasigi et al., 2019) | 24000 |
| Abstractive | ReCoRD (Wang et al., 2019) | 121000 |
| | DREAM (Sun et al., 2019) | 10197 |
| | QuaRTz (Tafjord et al., 2019) | 3864 |
| | Tweet-QA (Xiong et al., 2019) | 10692 |

Table 3: Relevant statistics of the QA dataset used to train Abstainer.

# A  Appendix

## A.1  QA Datasets for Training Abstainer

Publicly available QA datasets have been previously leveraged for generating synthetic data (Mekala et al., 2022b) in weakly (Mekala and Shang, 2020; Mekala et al., 2020) and minimally supervised settings (Mekala et al., 2021, 2022a). In this paper, we use multiple extractive and abstractive QA datasets to generate synthetic unanswerable samples and train Abstainer. The details about datasets are mentioned in Table 3.

## A.2  Experimental Settings

We use the OpenAI API `text-davinci-001` for GPT-3 and `code-davinci-002` for Codex. The Abstainer is fine-tuned on 411732 answerable and 435898 unanswerable samples. The batch size is 32 and each batch contains an equal number of answerable and unanswerable samples. We used 8 × NVIDIA Tesla V100 for our experiments.

## A.3  Frequently Asked Questions

**What is the scope of the presented ideas?** We believe our idea can be easily extended to any semantic parsing tasks involving natural language interfaces; we considered Task-oriented parsing as a first step because of its simpler representation. Through this work, we wanted to highlight an important real-life task that LLMs such as GPT3, Codex, and T0 underperform. Therefore, we believe this is useful and hope our work motivates more researchers to focus on this shortcoming.

**Why didn't you extract all syntactic phrases of a certain type in the tree for slot-value detection?** Syntactic phrase-based extraction of slot values requires users to manually enter rules (a.k.a. labeling functions) for each slot. When the number of slots increases (e.g. 74 in MTOP), it demands significant manual effort from users, which our paper aims to reduce. Moreover, such rules would generally predict many false positives/negatives, and classifying or identifying the appropriate ones accurately requires training data, which is not available in our zero-shot setting. Therefore, we compare against strong LLMs that are known for their impressive zero-shot performance such as GPT3, Codex, T0.

**The baseline LLMs such as GPT3 and Codex are not trained for semantic parsing. Wouldn't this make the performance improvement using ZEROTOP less significant?** We compare against the T0 model which is fine-tuned on several QA datasets like our Abstainer model. We cannot fine-tune the GPT3 and Codex models on these datasets separately. However, these instruction-tuned GPT3 and Codex are known to perform well on several question-answering & reading comprehension benchmarks (Robinson and Wingate, 2023). Therefore, we consider them as competitive baselines. In this paper, we show that these perform worse on zero-shot task-oriented parsing even when it is converted into a QA task, for which they are known to perform well. The reason behind their poor performance is because (1.) they are biased toward predicting labels common in the pre-training data, and (2.) they frequently hallucinate text for unanswerable questions. Through our work, we present this shortcoming, analyze the cause, and propose a method to fix it.

**Why did you choose one prompt per slot and not multiple?** We can possibly consider multiple prompts per slot and ensemble the predictions, which would intuitively boost the performance. However, multiple prompts per slot imply more annotations. Our motivation behind this work is to minimize the human annotations, thus we chose a single prompt per slot.