# OpenReview forum: "ZEROTOP: Zero-Shot Task-Oriented Semantic Parsing using Large Language Models"
_EMNLP/2023/Conference — EMNLP 2023 Main_

### Official Review · Reviewer_h26o · 2023-07-25

**Soundness:** 4

**Excitement:**

4: Strong: This paper deepens the understanding of some phenomenon or lowers the barriers to an existing research direction.

**Paper Topic And Main Contributions:**

This paper proposes a prompting method for semantic parsing in a zero-shot fashion. A language model is prompted to perform intent classification and slot detection, supporting an additional level of nesting, i.e. a nested intent. No training data is used: the model has access to schema information and the only manually crafted data are questions associated with slot names.

First, the LM is prompted to describe the intent of a user utterance. Then, a sentence encoder is used to compute the similarity between the generated description and the available intents. The most similar intent is selected. This works better than constraining generation to intent labels and computing generation likelihood, since probabilities favor words that occur frequently in the LM pretraining corpora.

Slot filling is performed by asking the LM to answer a question for each slot associated with the predicted intent. The question is manually crafted for each slot. Since not all slots have a corresponding value in any given user utterance, the authors train an abstainer model (T0-3B) on available QA data that contain the not-answerable annotation. This way, the LM can abstain from filling a slot.

If a predicted slot can have a nested intent, the abstainer is prompted with the possible slots of the nested intent. If one of those is filled with a value, the nested intent is added to the interpretation.

The paper shows that being able to decide when not to fill a slot with a fine-tuned model is very important since LLM will always try to generate/hallucinate values.

Generating descriptions for the intents, slightly improves the similarity between utterances and descriptions compared to utterances and intent labels.

**Questions For The Authors:**

Could you add the questions crafted for each slot to the appendix?

In this line of research it may be important to decide when not to parse an intent, i.e. decide when a user utterance is not relevant for the given schema. This way we could have a truly zero-shot agent capable of parsing only relevant inputs. Sure, it is a topic for a different paper, but it would be nice to read some considerations on that.

**Reasons To Accept:**

The paper shows that an LM can be used to parse sentences starting from a schema, and that it is important to add abstaining capabilities to the LM so that it knows when a slot is not present in a utterance.

This area of research is interesting since it enables parsing unseen schemas.

**Reasons To Reject:**

Experiments cover a single dataset.

Determining if a slot is fillable could potentially still be done by prompting the LM, avoiding fine-tuning altogether. It would probably require crafting a prompt for each slot, but would help in motivating the need of fine-tuning the model on data that it is not super-related to the parsing task.

**Reproducibility:**

4: Could mostly reproduce the results, but there may be some variation because of sample variance or minor variations in their interpretation of the protocol or method.

**Reviewer Confidence:**

4: Quite sure. I tried to check the important points carefully. It's unlikely, though conceivable, that I missed something that should affect my ratings.

---

> ### Author Rebuttal · Authors · 2023-08-28
>
> We thank the reviewer for the review. Below, we address the reviewer’s main concerns:
>
> __More test sets__
> We wanted to consider the dataset with general-purpose intents and slots that align with the LLM’s pre-trained data. Therefore, MTOP perfectly fits this criteria and we show that it is unable to abstain in such a familiar setting. For other datasets with complex representation such as CalFlow, we expect it to hallucinate more and it requires more involved work. We leave this for future work.
>
> __Abstaining via Prompting__
> We tried zero-shot prompting since it is a zero-shot setting. GPT3/Codex/T0-3B constrained in Table 2 refers to zero-shot prompting results for the slot with very intuitive questions. For example, *what is the music genre?* is the question for the MUSIC_GENRE slot. As mentioned in lines 226-229, the output is constrained to be either from the utterance or from their corresponding phrases indicating that the question cannot be answered. We observed LLM’s incapability of abstaining from prediction when there is no slot in a zero-shot setting. Therefore, we used auxiliary publicly available QA datasets to train specifically for abstaining. Individual hand-tuning of prompts might help a little, however, given the amount of manual effort in hand-tuning and the zero-shot setting, we believe fine-tuning on data from a different task is a better choice. We agree with your point that this would help in motivating our design.
>
> __Questions for slot__
> Yes, we will make these questions for the slots public.
>
> __Abstaining for intent__
> Yes, that’s a great suggestion. Thank you! Abstaining at the intent level is very important for a dialogue agent and it is widely studied as "Out-of-distribution detection". We leave this for future work.

---

### Official Review · Reviewer_i51D · 2023-08-05

**Soundness:** 4

**Excitement:**

4: Strong: This paper deepens the understanding of some phenomenon or lowers the barriers to an existing research direction.

**Paper Topic And Main Contributions:**

The paper explores the use of LLM for zero-shot semantic parsing, proposing the ZEROTOP method, which leverages large language models for zero-shot semantic parsing by decomposing the problem into a set of question-answering tasks. Experimental results on MTOP shows the advantages of the proposed method.

**Reasons To Accept:**

1. The paper is well written and easy to follow. The details of the method are clearly explained.
2. The proposed ZEROTOP that combines abstractive and extractive question-answering, along with fine-tuning on public QA datasets using negative samples, is well-motivated.
3. The paper provides empirical evidence of the effectiveness of the proposed method, showing promising results on the MTOP dataset.

**Reasons To Reject:**

1. Experiments are only conducted on a single dataset
2. The paper doesn't have a "conclusion" section, which makes the paper somehow incomplete.


**Reproducibility:**

4: Could mostly reproduce the results, but there may be some variation because of sample variance or minor variations in their interpretation of the protocol or method.

**Reviewer Confidence:**

3: Pretty sure, but there's a chance I missed something. Although I have a good feel for this area in general, I did not carefully check the paper's details, e.g., the math, experimental design, or novelty.

---

> ### Author Rebuttal · Authors · 2023-08-28
>
> We thank the reviewer for the review. Below, we address the reviewer’s main concerns:
>
> __More test sets__
> We wanted to consider the dataset with general-purpose intents and slots that align with the LLM’s pre-trained data. Therefore, MTOP perfectly fits this criteria and we show that it is unable to abstain in such a familiar setting. For other datasets with complex representation such as CalFlow, we expect it to hallucinate more and it requires more involved work. We leave this for future work.
>
> __Conclusion section__
> Due to the space limit, we removed the "Conclusion" section. We will add it back in the camera-ready version.

---

### Official Review · Reviewer_7287 · 2023-08-05

**Soundness:** 4

**Excitement:**

4: Strong: This paper deepens the understanding of some phenomenon or lowers the barriers to an existing research direction.

**Paper Topic And Main Contributions:**

This paper presents a novel method zero-shot semantic parsing using a fine-tuned LLM. The authors describe several issues with using prompting strategies for predicting intent and slot values from text using LLMs. First, the fact that the LLMs tend to hallucinate values for slot values missing from the text. Second, that LLMs tend to be strongly biased toward labels seen more frequently in training data when asked directly to predict an intent label.

1) to deal with the latter, the authors simply ask the model a general question for intent prediction, like "what did the user intend to do" and allow the model to freely generate a response. They then use a similarity scorer to match the generated text to the most similar intent label.

2) to deal with the former issue, they fine-tune an LLM to create an "abstainer" model which is specifically trained on questions to which no answer is present in the provided text. The model learns to abstain from answering such questions.

The authors pipeline involves first predicting intent using the method described in 1) and then repeatedly prompting the abstainer model to fill with questions associated with the intent's slot values to try to fill in those for which values are available. The authors write these questions for all of the slot values manually, somewhat adding to the overhead of using their proposed approach.

The authors show significant zero-shot parsing improvement over several competitive zero-shot semantic parsers in tests on a sample of the MTOP dataset.

**Reasons To Accept:**

The paper is clearly written, concise, and easy to follow.

The authors describe some interesting solutions to problems faced when using LLMs for zero-shot parsing (and other zero-shot labeling and recommendation tasks).

The authors show good results using their proposed method.

I think many in the community would find this paper interesting.

The paper shows the continued utility of some of the smaller LLMs in CL which I think some have lost focus of in the GPT frenzy.

**Reasons To Reject:**

The authors could have done more testing of their approach on alternate datasets, and could have tested using alternate models for their abstainer fine-tuning.

I think that this work could easily be restructured into a long paper with additional experiments. If the paper is not accepted, the authors should consider this.

Overall, though, these are minor considerations.

**Reproducibility:**

4: Could mostly reproduce the results, but there may be some variation because of sample variance or minor variations in their interpretation of the protocol or method.

**Reviewer Confidence:**

3: Pretty sure, but there's a chance I missed something. Although I have a good feel for this area in general, I did not carefully check the paper's details, e.g., the math, experimental design, or novelty.

---

> ### Author Rebuttal · Authors · 2023-08-28
>
> We thank the reviewer for the review. Below, we address the reviewer’s main concerns:
>
> __More test sets__
> We wanted to consider the dataset with general-purpose intents and slots that align with the LLM’s pre-trained data. Therefore, MTOP perfectly fits this criteria and we show that it is unable to abstain in such a familiar setting. For other datasets with complex representation such as CalFlow, we expect it to hallucinate more and it requires more involved work. We leave this for future work.
>
> __Long Paper Suggestion__
> Thank you for your suggestion! We appreciate it.

---

### Meta-Review · Area_Chair_5k3w · 2023-09-18

**Recommendation:** 5

**Metareview:**

This paper explores the use of LLMs for zero-shot semantic parsing where the main idea is to adapt the formulation of the task (semantic parsing) into an (extractive or abstractive) QA problem, which is a very natural thing to do. The LLMs also need the capability to figure out when a specific slot is missing and the authors show how it is possible to teach the model (i.e., tune it) that functionality via using synthetic negative examples. All the reviewers have detected a good deal of strengths of the paper, and I align with the overall sentiment of the reviewers.

The results are reported on a single dataset: MTOP, and I agree that the main issue of the current work is the fact that it evalutes on a single dataset, a wider coverage of other related semantic parsing problems and datasets would make the main findings more generalisable and more impactful eventually.

---

### Decision · Program_Chairs · 2023-10-07

**Decision:**

Accept-Main

**Comment:**

This paper explores the use of LLMs for zero-shot semantic parsing where the main idea is to adapt the formulation of the task (semantic parsing) into an (extractive or abstractive) QA problem, which is a very natural thing to do. The LLMs also need the capability to figure out when a specific slot is missing and the authors show how it is possible to teach the model (i.e., tune it) that functionality via using synthetic negative examples. All the reviewers have detected a good deal of strengths of the paper, and I align with the overall sentiment of the reviewers.

The results are reported on a single dataset: MTOP, and I agree that the main issue of the current work is the fact that it evalutes on a single dataset, a wider coverage of other related semantic parsing problems and datasets would make the main findings more generalisable and more impactful eventually.